# Pathologic Response Rates after Neoadjuvant Therapy for Sarcoma: A Single Institution Study

**DOI:** 10.3390/cancers13051074

**Published:** 2021-03-03

**Authors:** Crystal Seldon, Gautam Shrivastava, Melanie Fernandez, John Jarboe, Sheila Conway, Juan Pretell, Laura Freedman, Aaron Wolfson, Wei Zhao, Deukwoo Kwon, Andrew Rosenberg, Ty Subhawong, Jonathan Trent, Raphael Yechieli

**Affiliations:** 1Department of Radiation Oncology, Sylvester Comprehensive Cancer Center, University of Miami, Miami, FL 33136, USA; crystal.seldon@jhsmiami.org (C.S.); gautam.shrivastava@med.miami.edu (G.S.); m.fernandez68@umiami.edu (M.F.); JJarboe@carollc.com (J.J.); lfreedman@med.miami.edu (L.F.); awolfson@med.miami.edu (A.W.); 2Department of Orthopedic Surgery, University of Miami, Miami, FL 33136, USA; sconway@med.miami.edu (S.C.); j.pretell@med.miami.edu (J.P.); 3Biostatistics and Bioinformatics Shared Resource, Sylvester Comprehensive Cancer Center, University of Miami, Miami, FL 33136, USA; wzhao2@med.miami.edu (W.Z.); dkwon@med.miami.edu (D.K.); 4Department of Public Health Sciences, University of Miami, Miami, FL 33136, USA; 5Department of Pathology, University of Miami, Miami, FL 33136, USA; arosenberg@med.miami.edu; 6Department of Radiology, University of Miami, Miami, FL 33136, USA; tsubhawong@med.miami.edu; 7Department of Hematology Oncology, Sylvester Comprehensive Cancer Center, University of Miami, Miami, FL 33136, USA; jtrent@med.miami.edu

**Keywords:** soft tissue sarcoma, neoadjuvant therapy, pathologic complete response

## Abstract

**Simple Summary:**

For high-grade soft tissue sarcomas (STS), combined modality treatment with surgery and radiation therapy is the standard of care. The addition of chemotherapy has been shown to decrease the risk of local recurrence and improve survival. Evaluating treatment response with surrogate modalities such as MRI, CT and PET imaging have substantial limitations. Pathologic necrosis of the surgical specimen is a direct indicator of the effect of treatment on tumor cells. Studies in STS and other malignancies have shown that increasing rates of treatment-induced tumor necrosis correlate with improvement of oncological outcomes and survival. However, the relationship between pathologic response and outcomes of specific neoadjuvant treatments for STS remains indeterminate. We hypothesized that sequential neoadjuvant chemotherapy and radiation yields higher rates of pathologic complete response (pCR) than neoadjuvant radiation or chemotherapy alone. Our results indicate that neoadjuvant chemotherapy and radiation yields superior pCR compared to other neoadjuvant regimens.

**Abstract:**

(1) Background: Pathologic necrosis of soft tissue sarcomas (STS) has been used to determine treatment response, but its relationship to neoadjuvant treatments remains indeterminate. In this retrospective, single institution study, we hypothesized that neoadjuvant chemoradiation (NA-CRT) yields higher rates of pathologic complete response (pCR) than neoadjuvant radiation (NA-XRT) or chemotherapy (NA-CT) alone. (2) Methods: Patients with extremity STS between 2011–2020 who received neoadjuvant treatment were included. pCR was defined as percent necrosis of the surgical specimen greater than or equal to 90%. (3) Results: 79 patients were analyzed. 51.9% of the population were male with a mean age of 58.4 years. 49.4% identified as Non-Hispanic White. Twenty-six (32.9%) patients achieved pCR while 53 (67.1%) did not. NA-CT (OR 15.82, 95% CI = 2.58–96.9, *p* = 0.003 in univariate (UVA) and OR 24.7, 95% CI = 2.88–211.2, *p* = 0.003 in multivariate (MVA), respectively) and NA-XRT (OR 5.73, 95% CI = 1.51–21.8, *p* = 0.010 in UVA and OR 7.95, 95% CI = 1.87–33.7, *p* = 0.005 in MVA, respectively) was significantly associated with non- pCR when compared to NA-CRT. The analysis also demonstrated that grade 3 tumors, when using grade 2 as reference, also had significantly higher odds of achieving pCR (OR 0.23, 95% CI = 0.06–0.80, *p* = 0.022 in UVA and OR 0.16, 95% CI = 0.04–0.70, *p* = 0.015 in MVA, respectively). (4) Conclusion: NA-CRT yields superior pCR compared to other neoadjuvant regimens. This extends to higher grade tumors.

## 1. Introduction

Soft tissue sarcomas (STS) are rare tumors derived from mesenchymal cells [1]. STS involve less than 1% of all adult malignancies with an expected incidence of 13,130 new cases in the United States in 2020 [2]. Localized STS have a five-year relative survival rate of 81%, however the survival rate drops to 16% with distant spread of disease [3].

Treatment approaches for STS vary across sarcoma centers of excellence [4]. For high-grade tumors, combined modality treatment with surgery and radiation therapy is the standard of care [5], preferably with neoadjuvant radiation (NA-XRT) to reduce late toxicities of fibrosis, edema, and joint stiffness [6,7]. Although the addition of adjuvant chemotherapy to surgery with or without radiation decreases the risk of local recurrence and improves survival [8], this approach has not been globally implemented. Recently, neoadjuvant chemotherapy (NA-CT) and neoadjuvant sequential chemoradiation (NA-CRT) have been explored as treatment regimens. A large meta-analysis identified benefits in survival, distant recurrence (DR), and local recurrence (LR) with the addition of chemotherapy to localized therapy for STS. These benefits were further improved with the addition of ifosfamide to doxorubicin-based regimens [8]. Additional trials evaluating NA-CRT have shown benefits in overall survival (OS) and local control [9,10]. At our institution, patients with high-grade extremity sarcomas are treated with sequential neoadjuvant chemotherapy and radiation therapy followed by an oncologic resection. 

Surrogate modalities to measure tumor treatment response such as MRI, CT, and PET imaging, remain with substantial limitations. Pathologic necrosis of the surgical specimen is a direct indicator of the effect of treatment on tumor cells. Studies in STS and other malignancies have shown that increasing rates of treatment-induced tumor necrosis correlate with prognostic oncological outcomes and survival [11,12,13,14]. However, the relationship between pathologic response and outcomes of specific neoadjuvant treatments remains indeterminate. In this retrospective single institution study, we sought to determine whether sequential NA-CRT yields higher rates of pathologic response and superior oncologic outcomes than either NA-XRT or NA-CT alone. 

## 2. Materials and Methods

A retrospective review was conducted on 289 patients diagnosed with extremity STS between 2011 and 2020 at the University of Miami. This analysis was approved by the University of Miami Institutional Review Board 20190880 and due to the retrospective nature of this study, informed consent was waived. All patients were older than 18 and treated with NA-CT, NA-XRT, or NA-CRT followed by oncologic resection performed by the Orthopedic oncology team.

Exclusion criteria included patients who did not receive neoadjuvant therapy followed by an oncologic resection. Patients with low-grade sarcoma, rhabdomyosarcoma, extraosseous Ewing, primitive neuroectodermal tumors, osteosarcoma, chondrosarcoma, Kaposi’s sarcoma, angiosarcoma of the scalp/face, or any sarcoma of the head and neck were excluded based on RTOG 0630 [15] and RTOG 9514 [10] criteria. Neoadjuvant radiation therapy was delivered in 25 fractions to a dose of 50 Gy in 2 Gy per day increments. Resection was performed an average of 10 weeks after completion of radiation therapy.

Age was calculated from date of birth to date of tissue diagnosis. Survival status was taken from the medical record as was gender, race, and ethnicity. Patients identified their gender as male or female. Ethnicity was classified as Hispanic or non-Hispanic. For race, patients identified as White, Black, Asian, multiracial, or unknown. The date of diagnosis was the date of pathologic confirmation of STS. Tumor location was classified as originating in the upper or lower extremity. Tumor grade and histology were identified on the pre-operative pathology report. The following data were obtained from the surgical pathology report: percent necrosis, margin status, distance from closest margin, tumor size, treatment effect, and number of positive nodes. Status at last follow-up was categorized as alive without recurrence, alive with local regional recurrence, alive with distant recurrence, or deceased. All staging for analysis was based on the American Joint Committee on Cancer (AJCC) Staging System for STS of the trunk and extremities, 8th edition. Pathologic complete response (pCR) was defined as percent necrosis of the surgical specimen greater than or equal to 90% based on criteria from the ARST1321 trial [16]. Recurrences were listed as local if they occurred in the same region as the primary tumor or distant if they occurred outside of the primary location. Date of recurrence was listed as the date of pathological confirmation of recurrent disease or radiologic confirmation in the absence of a biopsy. Local recurrence free interval is defined as the time from surgery to the time of latest follow-up. 

Patient demographics and disease characteristics were summarized using descriptive statistics. For continuous variables, mean, standard deviation, median, minimum, and maximum values were reported. For categorical variables, frequencies were reported. Univariable and multivariable analyses (UVA and MVA) were performed to examine association between covariates and pCR using logistic regression. Stepwise variable selection was used for identifying independent variables in MVA. Overall survival analysis was performed using Cox proportional hazards regression for univariable analysis only due to small number of deaths. Kaplan Meier approach was used to evaluate survival curves among categorical groupings. Associations between distant recurrence and covariates were examined using Fine-Gray competing risks regression due to presence of competing risks (local recurrence and death without any recurrence). Pre- and post-treatment tumor size in centimeters (cm) was examined using logistic regression analysis. All statistical analyses were conducted using statistical software SAS version 9.4 and R version 3.6.3. Statistical significance was considered when *p* < 0.05.

## 3. Results

### 3.1. Clinical Characteristics

Clinical characteristics of the entire cohort by pathologic response are depicted in Table 1. Seventy-nine patients were included in the analysis and 51.9% of the population identified as male. The mean age of the cohort was 58.4 years. In addition, 49.4% of the cohort identified as Non-Hispanic White and 35.4% identified as Hispanic. Most STS were in the lower extremity (83.5%). There were many histologic subtypes represented in the cohort, however fibrosarcoma and fibroblastic sarcomas (57.0%) were the most common histology identified followed by synovial cell sarcomas (13.9%). Most tumors were grade 3 based on histologic interpretation (69.6%). Four patients had metastatic disease to the lung, however their disease was considered low volume. Two patients had nodal involvement on diagnosis. 

### 3.2. Treatment and Post-Therapy Response Chracteristics

Treatment and post-therapy characteristics of the entire cohort by pathologic response are depicted in Table 2. Twenty-six (32.9%) patients achieved pCR while 53 (67.1%) did not. Of those who received NA-CRT, 22 (55.0%) patients had tumor necrosis rates greater than 90% while 18 did not (45.0%). Of the NA-CT arm, 1 patient (5.0%) achieved pCR and 19 (95.0%) did not. Of the NA-XRT group, 3 patients achieved pCR (15.8%) while 16 (84.2%) did not. The median pre-treatment tumor size was 10.4 cm and post-treatment size was 9.7 cm based on radiographic interpretation. Of note, 66.7% of spindle cell sarcomas achieved pCR (Table 1). On comparing the pre- and post-treatment size of the tumors with paired T-tests, there was a significant difference among the patients that achieved pCR (*p* = 0.010) but not among the entire cohort (*p* = 0.161) or among patients who did not achieve pCR (0.997) (Figure 1). Pathology revealed negative margins for 70 patients (88.6%). 71 patients (89.9%) were alive at the time of the analysis. Of these 71 patients, 55 (69.6%) were without local or distant recurrence. Of the entire cohort, 23 patients developed a local or distant recurrence at the time of analysis. 19 (82.6%) of these patients did not achieve pCR. Seven of the eight patients who were deceased at the time of analysis had a local or distant recurrence at time of death. Of the patients who received chemotherapy, Adriamycin ifosfamide (AI) was the most common regimen used (34.2%).

### 3.3. Effect of Treatment on pCR

Logistic regression analysis was performed to determine the effect of treatment on pCR (Table 3). Variables examined were treatment, age, sex, race/ethnicity, location of primary tumor, histology, tumor size, and clinical T and N staging. On univariate and multivariate analysis, NA-CT (OR 15.82, 95% CI = 2.58–96.9, *p* = 0.003 in UVA and OR 24.7, 95% CI = 2.88–211.2, *p* = 0.003 in MVA, respectively) and NA-XRT (OR 5.73, 95% CI = 1.51–21.8, *p* = 0.010 in UVA and OR 7.95, 95% CI = 1.87–33.7, *p* = 0.005 in MVA, respectively) was significantly associated with non- pCR when compared to NA-CRT. Grade 3 disease was also significantly associated with achieving pCR in UVA when compared to Grade 2 disease (OR 0.23, 95% CI = 0.06–0.80, *p* = 0.022 in UVA and OR 0.16, 95% CI = 0.04–0.70, *p* = 0.015 in MVA, respectively). 

### 3.4. Predictors of OS and Recurrence

Univariable Cox models for OS were performed. (Table 4). No variables were found to be significantly associated with overall survival. The five-year OS rate for patients who achieved pCR was 100.0% compared to 81.2% for those who did not achieve pCR. Although there was a trend toward pCR correlation with superior OS when compared to non-pCR, the association was not significant, (*p* = 0.1809) (Figure 2). The five-year OS rates for patients who received NA-XRT, NA-CRT, and NA-CT were 100%, 77.9%, and 84.0%, respectively, which did not achieve statistical significance (*p* = 0.9353) as represented on Kaplan–Meier curve (Figure 3). At two years, the risk of cumulative distant recurrence was 20.7% (95% CI: 11.8–31.4%) (Figure 4).

## 4. Discussion

Within the different treatment groups, we determined that over half of those treated with NA-CRT (55.0%) achieved a level of pathologic necrosis of at least 90%, whereas in the groups with NA-CT and NA-XRT, only 5.0% and 15.8% achieved at least 90% necrosis, respectively. Our results demonstrate that the odds of achieving a pCR are significantly decreased with NA-CT (OR 15.82, 95% CI = 2.58–96.9, *p* = 0.003 in UVA and OR 24.7, 95% CI = 2.88–211.2, *p* = 0.003 in MVA, respectively) and NA-XRT (OR 5.73, 95% CI = 1.51–21.8, *p* = 0.010 in UVA and OR 7.95, 95% CI = 1.87–33.7, *p* = 0.005 in MVA, respectively) when compared to NA-CRT. The analysis also demonstrated that grade 3 tumors, when using grade 2 as reference, also had significantly higher odds of achieving pCR (OR 0.23, 95% CI = 0.06–0.80, *p* = 0.022 in UVA and OR 0.16, 95% CI = 0.04–0.70, *p* = 0.015 in MVA, respectively). These findings suggest that sequential neoadjuvant chemotherapy and radiation confer significantly greater degrees of pCRupon surgical resection as well as combined neoadjuvant therapy is effective for higher grade tumors compared to NA-CT or NA-XRT alone at achieving pCR. 

Prior studies have suggested NA-CRT is superior to solely preoperative radiotherapy, with NA-CRT increasing the likelihood of R0 resection [17]. A meta-analysis showed significantly greater five-year survival when comparing CRT to XRT (72.0% vs. 56.1%; *p* < 0.001) [18]. Smaller studies have contrasted survival outcomes between combined neoadjuvant chemotherapy/radiation and historical controls. One such study reported that the NA-CRT group had significantly better outcomes in survival (87% to 58%), disease-free survival (70% vs. 42%), and metastasis-free survival (75% vs. 44%) at 5 years [9]. Our study supports the record for NA-CRT by establishing its superiority to pre-operative chemotherapy or radiation therapy alone in the context of achieving pCR upon resection. 

The prognostic value of necrosis within a resected tumor has been the center of much debate as it continues being assessed in association with survival metrics. A study examining 113 individuals with sarcoma treated to achieve a median necrosis of 90% across samples found no difference in groups who had >95% necrosis and <95% necrosis in five-year follow-up for local control, disease-specific survival, and overall survival [19]. Other studies challenged the use of necrosis as a measure of treatment response entirely. In a study of 162 patients with STS treated with neoadjuvant therapy, a median follow-up of 4.5 years, and a median necrosis of 27%, the group found that higher rates of necrosis predicted worse outcomes for distant-metastasis-free, progression free, and OS [20,21]. On the contrary, the use of pCR across several other cancer types—including breast and head and neck cancer—is widely associated with improved outcomes including OS [22]. Our analyses corroborate the debate in the literature, showing that pCR as defined by greater than or equal to 90% necrosis was not significantly associated with OS when compared to non-pCR. 

Limitations of retrospective studies apply to our single center study. Chief among the limitations is the rarity of STS, accounting for the small sample size accrued over the nine-year time frame. Additionally, the lack of availability of comparable data across all 289 screened patients factored into the findings. Many patients went to different hospital systems for their surgeries and as a result did not have pathologic necrosis reported in their pathology and were excluded on that basis. Another limitation is the selection bias. Larger tumors (greater than 5 cm) are more likely to receive chemotherapy than smaller tumors due to higher risk of local and distant recurrence. In addition, patients with poor performance status are more likely to receive NA-XRT due to inability to tolerate chemotherapy. Additionally, challenges associated with follow-up led to exclusions due to incomplete records, limiting potential survival analyses. 

Our findings indicate that the use of sequential neoadjuvant chemotherapy and radiotherapy before surgical resection yields superior pCR. This finding extends to higher-grade tumors. Further prospective research into this approach will examine toxicity associated with greater therapeutic load, utilize the percentage decrease in tumor size in calculating percent necrosis, as well as apply correlative imaging through PET/CT or advanced MRI techniques to further quantify reductions in tumor burden and viability. Expanding the number of eligible patients for study could also add power to the study in determining whether pathologic complete response could serve as a surrogate for measures of survival outcomes. 

## 5. Conclusions

Sequential NA-CRT before surgical resection yields superior pCR when compared to NA-XRT and NA-CT. This finding extends to higher grade tumors. Although there was a trend toward pCR correlation with superior OS when compared to non-pCR, the association was not significant.

## Figures and Tables

**Figure 1 cancers-13-01074-f001:**
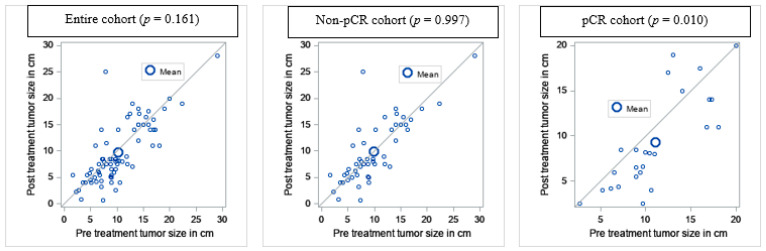
Comparing pre- and post-treatment tumor size in cm.

**Figure 2 cancers-13-01074-f002:**
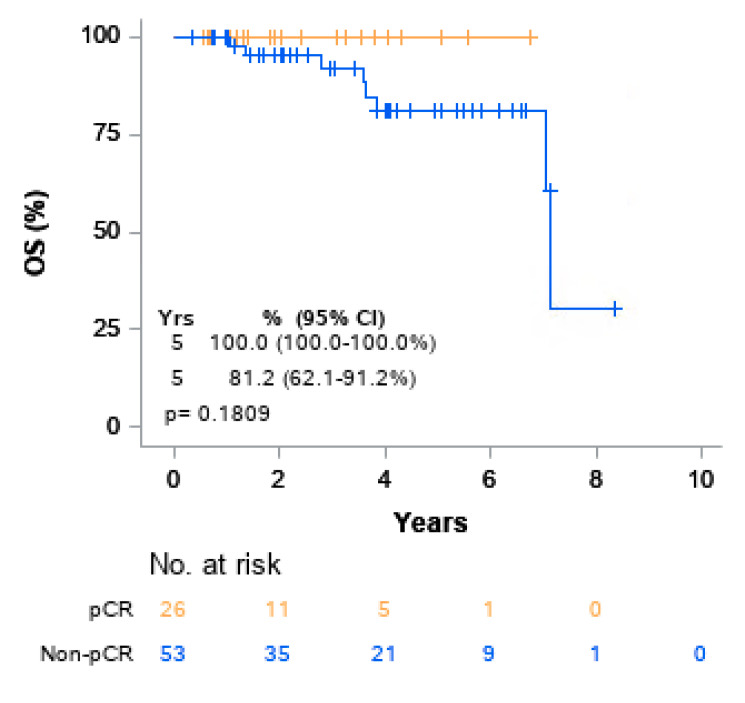
Kaplan–Meier curves of overall survival (OS) by pCR.

**Figure 3 cancers-13-01074-f003:**
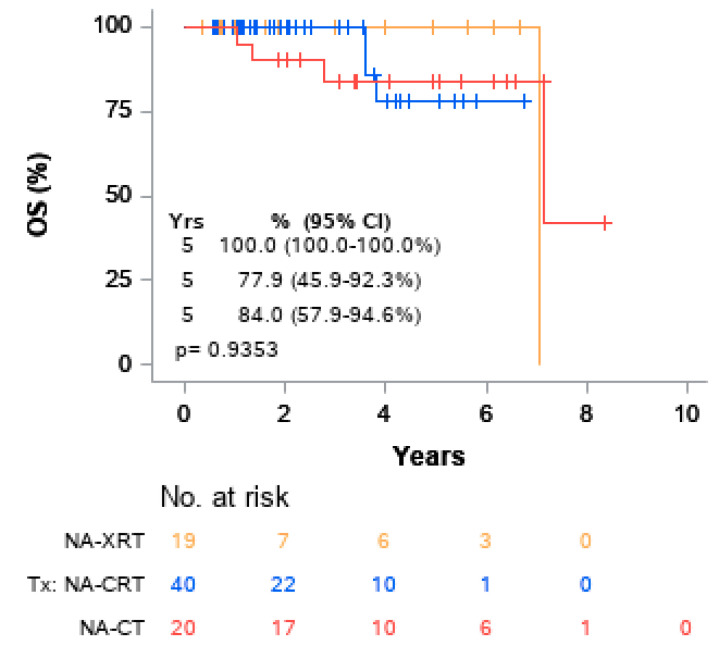
Kaplan–Meier curves of OS by treatment type.

**Figure 4 cancers-13-01074-f004:**
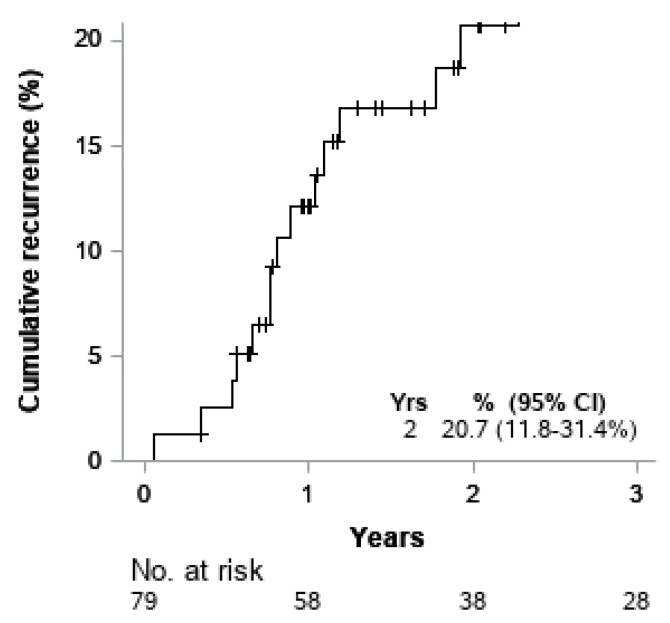
Cumulative incidence of distant recurrence.

**Table 1 cancers-13-01074-t001:** Baseline characteristic in all and by pCR.

Category	All		Pathologic Complete Response			
			pCR		Non-pCR	
	*N*	%	*N*	%	*N*	%
All	79	100.0	26	32.9	53	67.1
Gender						
Female	38	48.1	12	31.6	26	68.4
Male	41	51.9	14	34.1	27	65.9
Age						
Mean (SD)	58.4 (14.4)		59.4 (10.7)		57.9 (15.9)	
Median (Min, Max)	58.0 (23.0–83.0)		63.5 (32.0–75.0)		56.0 (23.0–83.0)	
Race/Ethnicity						
Non-Hispanic White	39	49.4	13	33.3	26	66.7
Non-Hispanic Black	10	12.7	4	40.0	6	60.0
Hispanic	28	35.4	8	28.6	20	71.4
Asian/Unknown	2	2.5	1	50.0	1	50.0
Location						
Upper extremity	13	16.5	4	30.8	9	69.2
Lower extremity	66	83.5	22	33.3	44	66.7
Histology						
Fibrosarcoma/Fibroblastic sarcoma	45	57.0	16	35.6	29	64.4
Leiomyosarcoma	6	7.6	1	16.7	5	83.3
Myxoid cell/round cell liposarcoma	5	6.3	2	40.0	3	60.0
Synovial cell sarcoma	11	13.9	1	9.1	10	90.9
Spindle cell sarcoma	6	7.6	4	66.7	2	33.3
Dedifferentiated liposarcoma	6	7.6	2	33.3	4	66.7
Grade						
2	24	30.4	3	12.5	21	87.5
3	55	69.6	23	41.8	32	58.2
Clinical T stage						
1	9	11.4	1	11.1	8	88.9
2	34	43.0	11	32.4	23	67.6
3	19	24.1	8	42.1	11	57.9
4	16	20.3	6	37.5	10	62.5
x	1	1.3	-	-	1	100.0
Clinical N stage						
0	76	96.2	26	34.2	50	65.8
1	2	2.5	-	-	2	100.0
x	1	1.3	-	-	1	100.0
Clinical M stage						
0	74	93.7	25	33.8	49	66.2
1	4	5.1	1	25.0	3	75.0
x	1	1.3	-	-	1	100.0
Clinical stage						
2	9	11.4	1	11.1	8	88.9
3	63	79.7	24	38.1	39	61.9
4	6	7.6	1	16.7	5	83.3
x	1	1.3	-	-	1	100.0

SD: standard deviation; Min = minimum; Max = maximum.

**Table 2 cancers-13-01074-t002:** Treatment and response characteristics.

Category	All		Pathologic Complete Response			
			pCR		Non-pCR	
	*N*	%	*N*	%	*N*	%
All	79	100.0	26	32.9	53	67.1
Neoadjuvant treatment						
NA-CRT	40	50.6	22	55.0	18	45.0
NA-CT	20	25.3	1	5.0	19	95.0
NA-XRT	19	24.1	3	15.8	16	84.2
Pre-treatment tumor size (cm)						
<5	10	12.7	1	10.0	9	90.0
5–10	36	45.6	12	33.3	24	66.7
>10	33	41.8	13	39.4	20	60.6
Mean (SD)	10.4 (5.0)		11.1 (4.4)		10.0 (5.4)	
Mean (Min, Max)	9.6 (1.7–29.0)		10.2 (2.7–20.0)		8.8 (1.7–29.0)	
Post-treatment tumor size (cm)						
<5	14	17.7	6	42.9	8	57.1
5–10	32	40.5	10	31.3	22	68.8
>10	29	36.7	9	31.0	20	69.0
Unknown	4	5.1	1	25.0	3	75.0
Mean (SD)	9.7 (5.7)		9.3 (5.3)		10.0 (6.0)	
Mean (Min, Max)	8.0 (0.6–28.0)		8.1 (2.5–20.0)		8.0 (0.6–28.0)	
Margins						
Negative	70	88.6	26	37.1	44	62.9
Positive	9	11.4	-	-	9	100.0
Chemotherapy						
AI	27	34.2	14	51.9	13	48.1
MAI	11	13.9	4	36.4	7	63.6
Adriamycin/Cytoxan	3	3.8	1	33.3	2	66.7
Adriamycin/Cisplatin	4	5.1	-	-	4	100.0
Other	15	19.0	4	26.7	11	73.3
Unknown	19	24.1	3	15.8	16	84.2
Cycles						
Mean (SD)	3.7 (1.4)		33.6 (1.3)		3.7 (1.4)	
Mean (Min, Max)	4.0 (1.0–7.0)		4.0 (1.0–6.0)		4.0 (2.0–7.0)	
Vital status						
Alive	71	89.9	26	36.6	45	63.4
Dead	8	10.1	-	-	8	100.0
Event of first failure						
Alive without recurrence	55	69.6	22	40.0	33	60.0
Local regional/distance recurrence	23	29.1	4	17.4	19	82.6
Dead without recurrence	1	1.3	-	-	1	100.0

NA-CRT: Neoadjuvant chemoradiation; NA-CT: Neoadjuvant chemotherapy; NA-XRT: Neoadjuvant radiation therapy; SD: Standard deviation; Min = maximum; Max = maximum; AI: Adriamycin ifosfamide; MAI: Mesna doxorubicin ifosfamide.

**Table 3 cancers-13-01074-t003:** Logistic regression: Effect of treatment on pCR vs. Non-pCR.

Variable	Category	UVA		MVA	
		OR (95% CI)	*p*-Value	OR (95% CI)	*p*-Value
Treatment	NA-CRT	Reference		Reference	
	NA-CT	15.8 (2.58, 96.9)	0.003	24.7 (2.88,211.2)	0.003
	NA-XRT	5.73 (1.51, 21.8)	0.010	7.95 (1.87, 33.7)	0.005
Age in years	One year increased	0.99 (0.96, 1.03)	0.678		
Sex	Female	Reference			
	Male	0.89 (0.35, 2.28)	0.816		
Race/ethnicity	Non-Hispanic White	Reference			
	Non-Hispanic Black	0.74 (0.18, 3.06)	0.673		
	Hispanic	1.23 (0.43, 3.51)	0.701		
	Asian/Unknown	0.51 (0.03, 8.80)	0.643		
Histology	Leiomyosarcoma	2.05 (0.27, 15.8)	0.491		
	Myxoid cell/round cell liposarcoma	0.78 (0.12, 5.13)	0.799		
	Synovial cell sarcoma	3.92 (0.59, 25.9)	0.156		
	Spindle cell sarcoma	0.31 (0.05, 1.84)	0.197		
	Dedifferentiated liposarcoma	1.01 (0.17, 5.96)	0.994		
Location	Lower extremity	Reference			
	Upper extremity	1.07 (0.30, 3.80)	0.920		
Pre-treatment tumor size in	<5	Reference			
cm	5–10	0.31 (0.04, 2.14)	0.234		
	>10	0.24 (0.03, 1.66)	0.148		
Post-treatment tumor size	<5	Reference			
in cm	5–10	1.64 (0.45, 5.97)	0.454		
	>10	1.65 (0.44, 6.15)	0.455		
	Unknown	1.78 (0.16, 19.37)	0.635		
Grade	2	Reference		Reference	
	3	0.23 (0.06, 0.80)	0.022	0.16 (0.04, 0.7)	0.015
Clinical T stage	1	Reference			
	2	0.36 (0.05, 2.57)	0.309		
	3	0.24 (0.03, 1.84)	0.169		
	4	0.29 (0.04, 2.30)	0.239		
	x	0.72 (0.00, 137.1)	0.901		
Clinical N stage	0	Reference			
	1	2.62 (0.06, 111.4)	0.614		
	x	1.65 (0.02, 164.9)	0.831		

NA-CT: Neoadjuvant chemotherapy; NA-CRT: Neoadjuvant chemoradiation; NA-XRT: Neoadjuvant radiation; NA: Not applicable. NE: Not estimable.

**Table 4 cancers-13-01074-t004:** Univariate Cox models for OS.

Variable	Category	HR (95%CI)	*p*
Treatment	NA-CRT	Reference	
Neoadjuvant treatment	NA-CT	0.49 (0.07,3.54)	0.476
	NA-XRT	0.59 (0.06,6.36)	0.667
Age	One year increased	0.99 (0.93,1.05)	0.745
Sex	Female	Reference	
	Male	0.12 (0.01,1.03)	0.054
Race/Ethnicity	Non-Hispanic White	Reference	
	Non-Hispanic Black	1.32 (0.14,12.2)	0.807
	Hispanic	0.35 (0.04,3.05)	0.342
	Asian/Unknown	NE	
Location	Lower extremity	Reference	
	Upper extremity	2.34 (0.42,12.9)	0.330
Pre-treatment tumor (cm)	<5	Reference	
	5–10	0.44 (0.06,3.11)	0.410
	>10	0.35 (0.04,2.92)	0.331
Post treatment tumor (cm)	<5	NE	
	5–10		
	>10		
	Unknown		
Histology	Fibrosarcoma/Fibroblastic sarcoma	Reference	
	Leiomyosarcoma	NE	
	Myxoid cell/round cell liposarcoma	NE	
	Synovial cell sarcoma	3.96 (0.65,24.3)	0.137
	Spindle cell sarcoma	NE	
Grade	2	Reference	
	3	0.56 (0.11,2.79)	0.476
Clinical T stage	1	Reference	
	2	0.37 (0.03,4.65)	0.440
	3	1.04 (0.09,12.7)	0.977
	4	0.32 (0.02,6.60)	0.462
	x	1.91 (0.11,33.0)	0.656
Clinical N stage	0	Reference	
	1	NE	
	x	3.63 (0.32,41.5)	0.299
Clinical M stage	0	Reference	
	1	NE	
	x	3.23 (0.30,34.7)	0.332
Clinical stage	2	Reference	
	3	0.41 (0.04,4.16)	0.452
	4	2.81 (0.16,50.3)	0.483
	x	1.92 (0.11,33.0)	0.652
Pathology complete response	pCR	NE	
(pCR)	Non-pCR		

NA-CT: Neoadjuvant chemotherapy; NA-CRT: Neoadjuvant chemoradiation; NA-XRT: Neoadjuvant radiation; NE: Not estimable.

## Data Availability

The data presented in this study are available on request from the corresponding author. The data are not publicly available due to patient privacy.

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
