# Peer review of "Pathologic Response Rates after Neoadjuvant Therapy for Sarcoma: A Single Institution Study"

_cancers, 2021, doi:10.3390/cancers13051074_

Round 1

Reviewer 1 Report

The manuscript described the pathologic response as neoadjuvant therapy to the soft tissue sarcoma (STS). The topic is interesting and important. But I have some concerns about the present study. 

1) Of 289 patients (line 67), why did 79 patients undergo neoadjuvant therapy? How many patients who received surgical excision alone are there? I would like to know the indication of the neoadjuvant therapy. Also, the authors should show the criteria of the neoadjuvant chemotherapy and/or radiotherapy.

2) When did the authors decide the treatment strategy, have histological type affected the decision making?

3) I think the authors should mention the detail of spindle cell sarcoma because 66.7% of the patients acquired pCR. Do you mean UPS?

4) How about the relationship between enhanced MRI findings and pathological findings? Necrosis may be detected on MRI.

5) I wonder if local radiotherapy is related to improving survival in addition to the local control. Please discuss it.

6) Did you analyze metastatic-free survival? It should be interesting if the development of metastasis is related to the rate of pathological response.

7) Please show the timing and dose of radiotherapy.

8) Please show the local complication after surgery.

9)The authors should add new findings in addition to the present conclusion. 

Author Response

We thank you for considering our manuscript entitled, “Pathologic Response Rates After Neoadjuvant Therapy for Sarcoma: A Single Institution Study,” for publication in Cancers. We thank the reviewer for taking the time to review our manuscript and for their comments and suggestions. We have addressed the reviewer comments below on a point-by-point basis. Our responses follow the reviewer comments while text changes are in quotes. Within the manuscript, the changes have been tracked and highlighted using red underlined text.

Reviewer 2 Report

Thank you very much for the opportunity to review the paper.

The paper „sought to determine whether sequential NA-CRT yields higher rates of pathologic response and superior oncologic outcomes than either NA-XRT or NA-CT alone.” The study addresses the important question which neoadjuvant therapy in STS induces the best clinical outcomes. It retrospectively examined this question by reviewing the clinical records of 79 patients with extremity STS between 2011 and 2020.

In general, the paper is well written and the structure and message are clear. The number of statistical tests and the methods used need to be justified.

Details:

31, 32: Please explain abbr. MVA und UVA.

43: no of cases: where?

45-46: Do the treatments “vary” or is there a “standard”? That is a bit confusing.

50: “improves survival”. Source missing.

98: MVA: Which variables were chosen and why? Do you expect confounding?

Methods:

Figure 1 is not explained in meth. as far as I can see.

In general, there is a whole lot of tests in the result section. Not all of them are described in the method section.

Logistic Regression: You run a log reg. with 79 cases (53 events of non-pCR (or are pCR the events?), 11 predictors. Group sizes min was 2. Please explain how under such circumstances a log. reg. is possible and discuss potential statistical risks. The OR are extremely high.

Results: Please disentangle methods and results.

119/120: That sentence is not clear. What means majority and how where the (not existing) differences determined and tested?

Discussion:

Please address the problem of multiple testing.

G2/ G3: The OR is 0.23 resp. 0.16. So, the chance of achieving a non-pCR is higher in G3. This is in contrast to the conclusion. Is there a reference error? If so, is there a rational for the better outcome in G3?

Survival Analysis: How many patients reached the 5 years and how many were censored (reasons?)?

Table 4: Here NA-CT and NA-XRT have HR below 1. So: NA-CT and NA-XRT are “trending” toward superior OS. Also, female – male HR 0.12, nearly significant. This is strange. I strongly suspect that the statistical tests do not work here.

In summary: I advise to reduce the number of statistical tests and to discuss the applicability of the used tests again thoroughly with a statistician. There are inconsistencies in the tables (OR of grading, and treatment). While the content of the paper is worthy of publication, there are a number of methodological issues that need to be addressed.

Author Response

(The authors gave the same response as above.)

Round 2

Reviewer 1 Report

I read the revised manuscript. But the authors did not find additional new findings. I think the result "the favorable response to pathological necrosis by the chemo-radiation therapy" is a very weak impact on real-world experience and we won't change the present strategy on soft tissue sarcoma. We would like to know the response to the preoperative treatment before definitive surgery. Pathological examination is performed after surgery. If we know the response before surgery, we may change the plan of the surgical margin.

Reviewer 2 Report

Thank you for the opportunity to review again and for the authors' responses. I have no further comments.